# Globalization and employment nexus: Moderating role of human capital

**Mansoor Mushtaq**[1], **Shabbir Ahmed**[2], **Mochammad Fahlevi**[3]*, **Mohammed Aljuaid**[4], **Sebastian Saniuk**[5]

**1** FAST School of Management, National University of Computer and Emerging Sciences, Islamabad, Pakistan, **2** Department of Economics, Government Islamia Graduate College, Kasur, Pakistan, **3** Management Department, BINUS Online Learning, Bina Nusantara University, Jakarta Barat, Indonesia, **4** Department of Health Administration, College of Business Administration, King Saud University, Riyadh, Saudi Arabia, **5** Department of Engineering Management and Logistic Systems, Faculty of Economics and Management, University of Zielona Góra, Zielona Góra, Poland

* mochammad.fahlevi@binus.ac.id

**Data Availability Statement:** The open access datasets employed in the analyses can be accessed from the following links: https://databank. worldbank.org/source/world-development-indicators (accessed on 15 June, 2022).

## Abstract

While globalization has increased the movement and interconnection of goods, technology, and information, it has also affected employment. Many studies have analyzed the impact of globalization on employment creation resulting in positive and negative findings. However, an area of literature still needs to be explored studying how human capital affects the impact of globalization on employment creation. The current study contributes to the literature by analyzing the moderating role of human capital in the globalization-employment nexus in 26 Asian countries. For this, annual panel data were collected from 1996 to 2019. The estimations have been done using 12 model specifications, 6 for direct and 6 for indirect impact association between globalization and employment through the human capital channel. The study uses generalized least square (GLS) method and generalized method of moments (GMM) for empirical analysis. The static and dynamic analysis shows that globalization's direct and indirect impact on employment through the channel of human capital is positive. Industrial value added and economic growth leads to more employment creation, whereas population growth dampens it. Human capital plays a positive role in getting the advantage of globalization in terms of employment creation. This study confirms the literature recommendations of promoting human capital development to achieve globalization's benefits for more employment creation.

## 1. Introduction

Globalization has significantly changed the international economy by spreading economic transactions beyond national borders [1]. It has posed significant challenges to the labor markets of emerging countries [2]. The influence of economic globalization on labor markets is often discussed in the literature using two basic methods. According to the optimistic viewpoint, encouraging investment and output in appropriate sectors in developing nations will help to enhance labor demand in these countries. In this approach, additional work

**Funding:** The authors would like to extend their appreciation to King Saud University for funding this work through the Researcher Supporting Project (RSP2022R481), King Saud University, Riyadh, Saudi Arabia. The funders had no role in study design, data collection and analysis, decision to publish, or preparation of the manuscript.

**Competing interests:** The authors have declared that no competing interests exist.

opportunities and an increase in labor income can be generated. Contrarily, some actions, such as reducing trade barriers emerging nations will implement to acquire a competitive advantage, may raise unemployment rates. According to this viewpoint, local businesses cannot compete with international businesses focused on technology-intensive production. As a result, countries rely more on imports instead of creating new job opportunities. Furthermore, macroeconomic changes caused by short-term capital flows may negatively affect employment security in these countries [3].

Globalization combines capital flows, foreign direct investment, trade, technology, and labor mobility across borders [4]. Economic strength, technology, social culture, and political issues play a role in this process. This demonstrates that globalization is not limited to a single country's economy. There are many components from different worlds. Over the last two decades, the globe has grown increasingly globalized as trade barriers have been reduced, technology has advanced rapidly, transportation and communication costs have decreased, and international migration has increased [5].

There is no consistency among researchers on the influence of globalization on employment. In terms of its influence on employment, globalization has advantages and drawbacks, as well as proponents and opponents. Corporations have developed tactics to take advantage of reduced tariffs, government subsidies and incentives, and an institutional framework that has grown to promote and facilitate the free movement of products and services across borders [6]. As firms moved processes offshore for reaping location advantages, work was generally assigned to regions with low costs, where governments offered better facilities and labor with better skill levels was available [7, 8]. This shift in the locus of investment and task performance for multinational corporations to increase profits has often resulted in declining employment in developed countries across various industries, beginning with low value-added manufacturing and gradually expanding to higher-skilled jobs in manufacturing as the services sector [5].

However, this is not the complete picture of continued job displacements and a scarcity of qualified applicants for newly generated employment. According to a contrary viewpoint, globalization, on the other hand, may assist underdeveloped countries directly and indirectly through cultural interaction, science, technology, trade, and finance. Following the reduction of global trade barriers, developing nations have benefited in terms of trade growth. Over the last few decades, literacy, enrolment, infant mortality, and life expectancy have all improved significantly. Companies and governments were compelled by international competition to cut labor expenses, which began to shrink as the unemployment rate grew [9]. Similarly, factors of production, especially skilled labor, have become relatively more mobile between countries as trade and foreign investments grow [10]. In less developed nations, globalization stimulates the reallocation of employment across a wide range of industries through trade, investment, and technology. Trade, among other things, is expected to accelerate the movement of employment from agriculture to the manufacturing and service sectors [11]. Therefore, the role of globalization is controversial and cannot be concluded without empirical analysis. The purpose of this study is to analyze this role for selected Asian economies.

According to Das and Ray [12], globalization and employment do not have long-run associations in most individual countries as well as in the panel of South Asian economies. They suggest that other channels might work between them as the domestic economic conditions sometimes determine employment. Any country's relatively better employment scenario implies that the prevailing working environment is supposed to be friendly, which might create a better business environment. Hence, foreign multinational corporations are likely to be induced to invest in these countries, leading to better employment opportunities. Secondly, Alfalih and Hadj [13] study the moderating role of human capital and institutions in the FDI-employment nexus in Saudi Arabia.

Following these two studies, the present study will cover at least two literature gaps. First, as pointed out by Alfalih and Hadj [13], it will study the moderating role of human capital in the globalization-employment nexus. Second, we are using a more comprehensive measure of globalization, i.e., the KOF index, as individual consideration of the sub-dimensions of globalization, i.e., foreign direct investment and trade liberalization, may be insufficient to explain the overall impact of globalization [13]. Second, they study the case of Saudi Arabia only. The present study will analyze the impact of a panel of twenty-six Asian economies.

In the remaining part of the study, section 2 will review recent literature. Section 3 will present methodology and data sources, and section 4 will discuss estimation results. Finally, section 5 will conclude.

## 2. Literature review

Although a large body of literature has proven the relevance and influence of globalization on aggregate employment and employment movements across sectors in wealthy nations, empirical data for developing countries is still lacking [11]. Different proxies of globalization, such as trade, FDI, and technological development, have been utilized in empirical research, making it impossible to compare studies directly [14–16]. These variables can, directly and indirectly, influence developing country labor markets [17]. The effect of globalization on employment rates is examined in three strands in the empirical literature. The first of these literature strands consider trade liberalization, and the second literature strand, foreign direct investment, is the primary indicator of economic globalization. The third strand measures globalization by using an index of globalization.

### 2.1. Nexus between trade and employment

Dutt, Mitra, and Ranjan [18] investigated the link between trade and unemployment and discovered that trade openness has an employment-enhancing effect. The panel data test results, on the other hand, suggest that trade liberalization has a short-term employment-increasing effect. Felbermayr, Prat, and Schmerer [19] used panel data and cross-sectional studies to investigate the impact of trade openness on unemployment rates. For the period 1980 to 2003, the panel data analysis covered 20 OECD countries. In contrast, the cross-sectional study covered 62 countries from 1990 to 2006. The findings demonstrate that increased trade openness lowers unemployment rates. Furthermore, the authors discovered that an influence that reduces unemployment leads to a new steady-state equilibrium in the long run [17].

Hasan et al. [20] looked at the link between trade liberalization and India's unemployment rate at the state and industry levels. The results suggest that trade liberalization has no negative impact on unemployment rates. Ogunrinola and Osabuohien [21], on the other hand, conclude that trade openness is a source of high unemployment in Nigeria's industrial sector from 1990 to 2006. Similarly, Yasmin and Khan [22] claim that trade liberalization has increased the elasticity of labor demand in manufacturing, magnifying any change in labor demand caused by increasing export volume. Their caveat is that better job prospects may be confined to highly trained workers. Mitra [23] examined the influence of trade on employment in the services sector using time series data. The study's findings suggest that trade does not substantially impact employment in India's services sector. Meidani and Zabihi [24] used Johansen-Juselius's cointegration analysis to look at the dynamic influence of globalization on unemployment rates in Iran from 1971 to 2006. In the study, the authors employ the trade intensity index (the ratio of total exports and imports to GDP) as a proxy for globalization. The findings show that globalization is a source of job creation in Iran. Malgouyres [25] looked at the effect of Chinese imports on France's sectoral employment change. According to a study, Chinese imports have transferred employment in France to the service industry. For a few selected

countries, Tarjáni [26] assessed the influence of trade on sectoral employment movements. The study discovered that trade is to blame for the shift in employment from agriculture to services in certain sample nations.

## 2.2. Nexus between FDI and employment

In order to attract foreign FDI, a variety of factors must be considered. It has the potential to deliver capital and technology to the enterprises and industries of receiving countries, therefore boosting domestic employment [27]. Seyf [28] used linear and nonlinear regression methods to look at the ability of foreign direct investment to create jobs in four European Union nations. According to the data, foreign direct investment has no meaningful influence on employment creation in these countries. Jenkins [29] looked at the direct and indirect impacts of foreign direct investment on Vietnam's unemployment. The findings show that this country's ability to create jobs, both directly and indirectly, is limited.

In Pakistan, Malik, Chaudhry, and Javed [30] discovered that FDI improves employment, whereas trade has a negative impact. Tang [31] used ARDL to investigate the causal link between inward FDI and domestic employment in Singapore's manufacturing and services sectors, finding that inward FDI improves employment in the manufacturing sector over time. Because these industries are mutually beneficial, employment in the services sector may supplement those in manufacturing. Similarly, Habib and Sarwar [32] discovered that FDI positively influenced overall employment. More specifically, boosting domestic employment through the growth of manufacturing facilities requires new labor for every new investment project, regardless of investors' origin [33]. Similarly, Zeb, Qiang, and Sharif [34] used the OLS approach to assess the influence of FDI on Pakistan's unemployment rate from 1995 to 2011. FDI has a significant role in reducing unemployment in Pakistan, according to the study.

Rizvi and Nishat [35], on the other hand, looked at the link between foreign direct investment and employment in China, India, and Pakistan. In all three countries examined, there was no statistically significant link between FDI and employment. According to Brincikova and Darmo [36], the employment impact of inward FDI depends on how it enters the host nation. Mergers and acquisitions would hurt employment, whereas greenfield investment would have a positive effect. However, they could not find empirical evidence of a substantial beneficial effect on the V4 countries between 1993 and 2012.

In order to adjust for similar labor market features and institutions, Marelli et al. [33] divided the EU regions into four portions as dummy variables. They developed a new explanatory variable to capture the FDI-induced indirect effects on employment. While the overall positive employment effect in EU areas is low, the indirect employment effect is significant and positive, according to the data. Meanwhile, Jude and Silaghi [37] used a dynamic labor demand model to evaluate panel data from 20 Central and Eastern European nations from 1995 to 2012. They show that because of labor-saving measures, FDI inflows have an initial negative employment effect. The gradual vertical integration of foreign affiliates into the local economy finally converges on a long-term beneficial impact in only EU nations. Later, Lee and Park [38] used firm-level data from 20 Korean industries to demonstrate that inbound greenfield FDI boosts industry employment. Similarly, Saucedo et al. [39] analyzed quarterly panel data from 32 Mexican states from 2005 to 2018, indicating that FDI had a favorable influence on manufacturing employment but no noticeable impact on service sector employment.

## 2.3. Nexus between globalization and employment

The third strand of literature considers globalization's economic, political, and social dimensions (published by the KOF globalization index) and tries to explain the relationship between

these measures and employment. Analyzing the impact of globalization on employment in Bangladesh and Kenya, Sen [40] found a positive relationship in Bangladesh. However, the study found a negative relationship between globalization and employment in Kenya.

According to Osmani [41], globalization has enhanced job possibilities in Bangladesh through a net increase in employment in the tradable goods industries and indirectly through increased demand for items produced in the non-tradable sector. In a similar vein, Majumder [42] claims that while globalization may have resulted in a higher per capita GDP in a developing economy, the benefits are unlikely to be shared equally throughout society, as evidenced by rising income inequality and a decline in the quality of employment opportunities outside of the modernized sector. Lee et al. [43] argue that increased globalization leads to lower unemployment in the urban sector. Malik et al. [30] conclude that economic globalization is linked to increased job prospects in Pakistan. Similarly, Gozgor [44] finds that the KOF globalization index significantly reduces unemployment in G7 economies. Awad and Youssof [45] examined the impact of the economic globalization index on unemployment in Malaysia, and the findings show that economic globalization has a decreasing impact on unemployment in Malaysia.

The improving role of economic globalization on employment has also been confirmed empirically by Daly et al. [46] for Pakistan, Gozgor [47] for 87 countries, Adamu, et al. [48] for 35 Sub-Saharan African countries, Siddiqa et al. [49] for developing countries. Sana et al. [11] empirically investigated the impact of globalization on employment shifts in the labor market of Pakistan for the period 1991–2017. The results indicated that sectoral shift in employment to the services sector is positively affected by globalization measured by trade openness and foreign direct investment.

According to Hossain et al. [50], globalization increases female labor force participation by generating new employment opportunities. However, the positive benefits are more in low and middle-income economies than in high-income economies. However, Roll, Semyonov, and Mandel [51] argue that, while globalization boosts women's labor-force involvement, it lowers women's relative chances of attaining profitable, high-status employment. The review of the literature shows that the results are inconclusive. Secondly, existing studies consider a direct association between globalization and employment. The current study covers these gaps by considering the moderating role of human capital in the relationship between globalization and employment.

The review of literature points towards two hypotheses regarding the relationship between globalization and employment creation:

H1: Globalization has a positive role in creating employment.

H2: Human capital plays a positive role in determining the impact of globalization on employment.

## 3. Theoretical framework, methodology, and data

According to the Heckscher-Ohlin model, globalization will increase employment in developing countries. Ohlin argued in the theory of relative comparative advantages that FDI and trade benefited from a labor surplus and supported the trend of specialization and expertise in labor at the local level [52]. However, according to Muhammad, Islam, and Bashir [53], foreign direct investment and trade do not increase employment in developing economies.

Globalization produces employment, but it may also eliminate them. According to the neo-liberal school, globalization is a ubiquitous "creative destruction" force in global, cross-border trade. Even though old jobs are being replaced and salaries for unskilled employees are decreasing dramatically, technological investments and innovations boost efficiency and

performance, resulting in unprecedented affluence [54]. Globalization, in this view, draws and produces a slew of dangers. Relevant empirical research has also demonstrated that globalization stimulates the expansion and development of the industrial sector in underdeveloped nations, hence lowering global income disparity [55, 56].

However, globalization increases business competitiveness, resulting in company closures, relocation to other countries, and job losses. Positive effects of globalization include internationalization of production due to companies with global operations; rapid assimilation of new technologies; privatization gaining global traction; telecommunications eliminating distances and physically bringing people closer together, ensuring awareness of global issues; financial and commercial markets entering a phase of integration in their activity and functioning; and encouraging political and economic natural regeneration. Globalization is advantageous from a variety of perspectives. Private enterprise is capable of generating income for the state. Global competition has liberated entrepreneurial and creative abilities while also hastening technical innovation [54, 57, 58].

### 3.1. Model specification

Under this information, the models to be applied in the study are given in Eqs 1 and 2, respectively. In terms of model specification, it has benefited from the studies of Alfalih and Hadj [13], Awad and Youssof [45], Daly et al. [46], and Gozgor [47]. However, the current study uses twelve model specifications to study globalization's direct and indirect impact on employment through the channel of human capital.

The general specification of the function can be written as:

$$\mathbf{EMP} = (\mathbf{IVA}, \textit{EG}, \mathbf{POP}, \mathbf{GLOB}) \tag{A}$$

$$\mathbf{EMP} = (\mathbf{IVA}, \textit{EG}, \mathbf{POP}, \mathbf{GLOB} * \mathbf{HC}) \tag{B}$$

In the econometric model, this can be expressed as:

$$\mathbf{EMP1} = \boldsymbol{\beta}01 + \boldsymbol{\beta}11\mathbf{IVA} + \boldsymbol{\beta}21\textit{EG} + \boldsymbol{\beta}31\mathbf{POP} + \boldsymbol{\beta}41\mathbf{GLOB}_1 + \boldsymbol{\mu} \tag{1}$$

$$\mathbf{EMP2} = \boldsymbol{\beta}02 + \boldsymbol{\beta}12\mathbf{IVA} + \boldsymbol{\beta}22\textit{EG} + \boldsymbol{\beta}32\mathbf{POP} + \boldsymbol{\beta}42\mathbf{GLOB}_2 + \boldsymbol{\mu} \tag{2}$$

$$\mathbf{EMP3} = \boldsymbol{\beta}03 + \boldsymbol{\beta}13\mathbf{IVA} + \boldsymbol{\beta}23\textit{EG} + \boldsymbol{\beta}33\mathbf{POP} + \boldsymbol{\beta}43\mathbf{GLOB}_3 + \boldsymbol{\mu} \tag{3}$$

$$\mathbf{EMP4} = \boldsymbol{\beta}04 + \boldsymbol{\beta}14\mathbf{IVA} + \boldsymbol{\beta}24\textit{EG} + \boldsymbol{\beta}34\mathbf{POP} + \boldsymbol{\beta}44\mathbf{GLOB}_4 + \boldsymbol{\mu} \tag{4}$$

$$\mathbf{EMP5} = \boldsymbol{\beta}05 + \boldsymbol{\beta}15\mathbf{IVA} + \boldsymbol{\beta}25\textit{EG} + \boldsymbol{\beta}35\mathbf{POP} + \boldsymbol{\beta}45\mathbf{GLOB}_5 + \boldsymbol{\mu} \tag{5}$$

$$\mathbf{EMP6} = \boldsymbol{\beta}06 + \boldsymbol{\beta}16\mathbf{IVA} + \boldsymbol{\beta}26\textit{EG} + \boldsymbol{\beta}36\mathbf{POP} + \boldsymbol{\beta}46\mathbf{GLOB}_6 + \boldsymbol{\mu} \tag{6}$$

The model, including interaction terms of globalization and human capital, is as follows:

$$\mathbf{EMP7} = \boldsymbol{\beta}07 + \boldsymbol{\beta}17\mathbf{IVA} + \boldsymbol{\beta}27\textit{EG} + \boldsymbol{\beta}37\mathbf{POP} + \boldsymbol{\beta}47\mathbf{GLOB}_1 * \mathbf{HC} + \boldsymbol{\mu} \tag{7}$$

$$\mathbf{EMP8} = \boldsymbol{\beta}08 + \boldsymbol{\beta}18\mathbf{IVA} + \boldsymbol{\beta}28\textit{EG} + \boldsymbol{\beta}38\mathbf{POP} + \boldsymbol{\beta}48\mathbf{GLOB}_2 * \mathbf{HC} + \boldsymbol{\mu} \tag{8}$$

$$\mathbf{EMP9} = \boldsymbol{\beta}09 + \boldsymbol{\beta}19\mathbf{IVA} + \boldsymbol{\beta}29\textit{EG} + \boldsymbol{\beta}39\mathbf{POP} + \boldsymbol{\beta}49\mathbf{GLOB}_3 * \mathbf{HC} + \boldsymbol{\mu} \tag{9}$$

$$\mathbf{EMP10} = \boldsymbol{\beta}010 + \boldsymbol{\beta}110\mathbf{IVA} + \boldsymbol{\beta}210\textit{EG} + \boldsymbol{\beta}310\mathbf{POP} + \boldsymbol{\beta}410\mathbf{GLOB}_4 * \mathbf{HC} + \boldsymbol{\mu} \tag{10}$$

$$EMP11 = \boldsymbol{\beta}011 + \boldsymbol{\beta}111\mathbf{IVA} + \boldsymbol{\beta}211\boldsymbol{EG} + \boldsymbol{\beta}311\mathbf{POP} + \boldsymbol{\beta}411\mathbf{GLOB}_5 * \mathbf{HC} + \boldsymbol{\mu} \quad (11)$$

$$EMP12 = \boldsymbol{\beta}012 + \boldsymbol{\beta}112\mathbf{IVA} + \boldsymbol{\beta}212\boldsymbol{EG} + \boldsymbol{\beta}312\mathbf{POP} + \boldsymbol{\beta}412\mathbf{GLOB}_6 * \mathbf{HC} + \boldsymbol{\mu} \quad (12)$$

We use annual panel data for twenty-six Asian courtiers from 1996 to 2019 (see S1 Appendix). The countries have been chosen based on data availability for all variables. The study uses data for Asian economies only due to their matching human capital and employment structure. The dependent variable is employment (EMP), measured by employment as a population percentage. This proxy is previously used by [37, 59, 60].

The first independent variable is an industrial value added as a percentage of GDP (IVA). This measure is previously used by [59, 61]. The second independent variable is economic growth measured by the natural logarithm of gross domestic product in current US dollars (EG). This measure is previously used by [60, 62, 63]. The third independent variable is population growth measured by the natural logarithm of total population (POP). This measure is previously used by [64]. The fourth independent variable is globalization (GLOB), measured by six proxies of the KOF index of globalization. Some recent studies explaining globalization's role in employment creation are [5, 6]. The human capital variable has been used as a moderator to show the indirect link between globalization and employment through the channel of human capital measured by the human capital index. Alfalih and Hadj [13] consider the role of human capital as a moderator of globalization-employment association. Based on the discussion of these mentioned studies, the expected sign of the coefficient of industrial growth and economic growth is positive. In contrast, the expected sign of the coefficient of population growth is negative.

The data on employment and the human capital index has been collected from the website of FRED. In contrast, data on all variables have been collected from the World Bank's world development indicators database. The study uses 12 model specifications based on six measures of globalization to examine the role of globalization in employment creation by using six proxies of globalization. The first six specifications have been used to analyze the direct impact. The following six specifications have been used to check the indirect impact of globalization on employment creation through the channel of human capital.

## 3.2. Methodology

The general form of the model to be estimated is given below:

$$Y_{i,t} = Z_{i,t}\beta + H_i\alpha + \varepsilon_{i,t} \quad (A)$$

i = cross-section dimension, t = time series dimension

$Y_{i,t} =$ Dependent variable of $i^{th}$ cross-section in $t^{th}$ time. $Z_{i,t} =$ Set of Independent variables.

$H_i =$ The heterogeneity, or discrete impact. It should be remembered that $H_i$ has an intercept term and a group of cross-section-specific variables. That group of variables may or may not be observed. If we suppose all these individualities are identified and obstinate, then this model forms a simple Classical Linear Regression Model (CLRM).

The following is the general form of the model to be estimated:

$$Y_{i,t} = Z_{i,t}\beta + H_i\alpha + \varepsilon_{i,t} \quad (B)$$

I = cross-section dimension, t = time series dimension,

Yi,t = I$^{th}$ cross-section dependent variable in t$^{th}$ time period. Set of independent variables (Zi,t). The distinct influence, or heterogeneity (Hi). It is important to recall that Hi has an

intercept term and a set of cross-sectional variables. That set of variables might be observed or not. Suppose we assume that all of these individuals have been identified and are stubborn. In that case, we have a straightforward Classical Linear Regression Model (CLRM).

Only the Ordinary Least Square (OLS) approach may be used to examine it econometrically. In reality, finding such a perfect condition is few and far between. When Hi is not specified, we usually see a wide range of positions. Hi has a connection to Ki,t in this circumstance. Because the model includes missing variables, OLS results are skewed and inconsistent in this situation. However, in this case, the model has the following shape:

$$Y_{it} = Z_{it}\beta + \alpha_i + \varepsilon_{it} \tag{C}$$

Where i = Hi has all the observed effects and computes an estimable conditional mean, in this case, the Fixed Effect (FE) Model is appropriate for an empirical estimate. The Fixed Effect Model considers I to be the regression's cross-section-specific intercept. When using Fixed Effect, we assume that anything in the cross-section affects or biases the explanatory or explained variable, which must be addressed. In light of the hypothesis of a link between individual error terms and explanatory factors, this is the reason. The net influence of the regressors on the dependent variable may be calculated. On the other hand, if we assume that undiscovered cross-section specific effects are unrelated to the independent variables, we arrive at the random Effect Model, which may be written as follows:

$$Y_{i,t} = Z_{i,t}\beta + E[H_i\alpha] + \{H_i\alpha - E[H_i\alpha]\} + \varepsilon_{i,t}$$
$$Y_{i,t} = Z_{i,t}\beta + \alpha + \mu_i + \varepsilon_{i,t} \tag{D}$$

It states that i is a country-specific random component like I,t in a Linear Regression Model with a compound disturbance term. Heteroskedasticity is typically a concern with this type of model. Diagnostic tests for endogeneity and over-identifying limitations will be invalid in this case. Using heteroskedasticity-consistent or "robust" standard errors and statistics can help alleviate these problems to some extent. Hausman [65] devised the Hausman specification test to determine whether to use the fixed effect model or the random effects model. The null hypothesis is that the random effects model is acceptable for empirical model estimation. In contrast, the alternative hypothesis is that the fixed effect model is adequate.

One fundamental econometric difficulty in empirical research is measurement inaccuracy. Errors-in-variables or errors-in-regressors are two types of measurement errors. The assumptions and estimate methods of ordinary least squares fail due to measurement inaccuracy. The coefficients of the OLS estimators are not efficient or unbiased in the presence of measurement error. Fuller [66] proposed a solution to the measurement inaccuracy problem. He argued that knowing the measurement error covariance matrix is required to estimate the model. They also support adding additional confirmation or repetition data to estimate the model as an alternative. In panel data, however, each cross-section provides many observations, which may be utilized as "partial" duplicates to keep measurement error under control. Griliches and Hausman [67] suggested that accurate estimators may be produced without knowing the measurement error covariance matrix or additional confirmation or repetition data in specific panel data models. In the circumstance mentioned above, a different approach known as the "Dynamic Panel Model" can be employed in panel research. The following is an example of a panel dynamic model extension:

$$X_{I,t} = Z_{i,t}\beta + PX_{i,\ t-1} + c_{i,t} \tag{E}$$

Because measurement error or missing variables might produce "Endogeneity" in the model, this sort of model can be evaluated using the Generalized Method of Moments [68]. In

panel data studies, GMM is a widely used approach [69]. They are generic estimators for situations where explanatory variables are not strictly exogenous. We deal with issues like heteroskedasticity and autocorrelation within a single entity [70]. GMM estimators are designed to estimate panel data under specific circumstances. The following are some of these conditions: (1) The number of cross-sections in the panel must be greater than the number of periods in the panel (N>T); (2) The explained or dependent variable should be used as a lagged dependent variable; (3) Explanatory variables are usually correlated with previous values, and possibly current realizations of the error; (4) Fixed personal effects are an essential assumption; and (5) Heteroskedasticity and autocorrelation within entities but not across them.

In the presence of heteroskedasticity, GMM offers efficient estimates. Due to its wide variety of applications, GMM has become a popular tool among empirical academics [71]. Effective GMM has the advantage of consistency in the presence of random heteroskedasticity but at the cost of a potentially small sample size. GMM estimators are frequently used to correct bias caused by endogenous variables. Because one of the explanatory variables is also part of explained variable, there may be an issue of "Endogeneity" in the model. We used the GMM system proposed by Blundell and Bond [71]. GMM system has the upper hand over GMM differences [72].

## 4. Results and discussion

Table 1 presents the descriptive statistics of the variables. The values show a considerable variation between minimum and maximum values. It also indicates that the values are spread out over an extensive range of values, as shown by the standard deviation.

### 4.1. Econometric analysis

As we used panel data, we had to decide whether the fixed or random effect model should be used. The Hausman test helps choose between the fixed and random effect models. We applied the Hausman test on the null hypothesis that the random effect model is appropriate for our data set. The results of the Hausman test are presented in Tables 2 and 3 below.

Tables 2 and 3 indicate that the null hypothesis is rejected for only two specifications. Therefore, two specifications in both models suggest a fixed effects model. Therefore, the Hausman test suggests using the fixed effects model for static analysis. However, the diagnostic tests shown in Tables 4 to 7 point out that the model suffers from heteroscedasticity and serial

**Table 1. Descriptive statistics.**

|  | Mean | Standard Deviation | Minimum | Maximum |
|---|---|---|---|---|
| **Employment** | 55.473 | 12.458 | 31.367 | 88.221 |
| **Industry Value Added** | 4.751 | 8.613 | -38.884 | 111.379 |
| **Economic Growth** | 25.580 | 1.872 | 20.779 | 30.289 |
| **Population** | 17.191 | 1.982 | 12.626 | 21.058 |
| **Globalization 1** | 59.689 | 11.231 | 33 | 84 |
| **Globalization 2** | 58.184 | 13.794 | 29 | 91 |
| **Globalization 3** | 61.168 | 9.891 | 35 | 84 |
| **Globalization 4** | 56.748 | 16.101 | 18 | 95 |
| **Globalization 5** | 56.491 | 19.868 | 17 | 99 |
| **Globalization 6** | 56.814 | 14.940 | 17 | 92 |
| **Human Capital Index** | 2.438 | 0.566 | 1.107 | 4.351 |

Source: Authors' calculations

**Table 2. Hausman test results for direct impact.**

**Null Hypothesis: Random effect model is appropriate**

| Regressions | Chi-square statistic | Chi-Sq. D.F | Probability |
|---|---|---|---|
| Specification 1 | 9.83 | 4 | 0.0435** |
| Specification 2 | 7.20 | 4 | 0.1259 |
| Specification 3 | 12.03 | 4 | 0.0171* |
| Specification 4 | 7.01 | 4 | 0.1352 |
| Specification 5 | 6.26 | 4 | 0.1806 |
| Specification 6 | 6.86 | 4 | 0.1436 |

\* and \*\* show significance levels at 1% and 5%, respectively.

Source: Authors' calculations

**Table 3. Hausman test results for indirect impact through human capital.**

**Null Hypothesis: Random effect model is appropriate**

| Regressions | Chi-square statistic | Chi-Sq. D.F | Probability |
|---|---|---|---|
| Specification 1 | 5.98 | 4 | 0.2009 |
| Specification 2 | 3.20 | 4 | 0.5254 |
| Specification 3 | 10.61 | 4 | 0.0313** |
| Specification 4 | 6.98 | 4 | 0.1369 |
| Specification 5 | 4.36 | 4 | 0.3601 |
| Specification 6 | 9.85 | 4 | 0.0430** |

\* and \*\* show significance levels at 1% and 5%, respectively.

Source: Authors' calculations

**Table 4. Wald test for heteroscedasticity (direct impact).**

**Null Hypothesis: Homoscedasticity**

| Regressions | Chi-square statistic | Probability |
|---|---|---|
| Specification 1 | 2891.87 | 0.0000* |
| Specification 2 | 4733.36 | 0.0000* |
| Specification 3 | 2972.70 | 0.0000* |
| Specification 4 | 3886.50 | 0.0000* |
| Specification 5 | 5683.44 | 0.0000* |
| Specification 6 | 3506.22 | 0.0000* |

\* and \*\* show significance levels at 1% and 5%, respectively.

Source: Authors' calculations

correlation problems. Therefore, it is better to employ the generalized least squares (GLS) method instead of the fixed effects model for static analysis.

## 4.2. Diagnostic tests

Tables 4–7.

**Table 5. Wooldridge test for serial correlation (direct impact).**

**Null Hypothesis: No Serial Correlation**

| Regressions | F-statistic | Probability |
|---|---|---|
| Specification 1 | 131.239 | 0.0000* |
| Specification 2 | 133.159 | 0.0000* |
| Specification 3 | 129.856 | 0.0000* |
| Specification 4 | 131.771 | 0.0000* |
| Specification 5 | 134.219 | 0.0000* |
| Specification 6 | 131.176 | 0.0000* |

* and ** show significance levels at 1% and 5%, respectively.

Source: Authors' calculations

**Table 6. Wald test for heteroscedasticity (indirect impact).**

**Null Hypothesis: Homoscedasticity**

| Regressions | Chi-square statistic | Probability |
|---|---|---|
| Specification 1 | 3988.98 | 0.0000* |
| Specification 2 | 6682.34 | 0.0000* |
| Specification 3 | 2785.55 | 0.0000* |
| Specification 4 | 4126.25 | 0.0000* |
| Specification 5 | 6086.29 | 0.0000* |
| Specification 6 | 3277.28 | 0.0000* |

* and ** show significance levels at 1% and 5%, respectively.

Source: Authors' calculations

**Table 7. Wooldridge test for serial correlation (indirect impact).**

**Null Hypothesis: No Serial Correlation**

| Regressions | F-statistic | Probability |
|---|---|---|
| Specification 1 | 136.698 | 0.0000* |
| Specification 2 | 134.548 | 0.0000* |
| Specification 3 | 135.977 | 0.0000* |
| Specification 4 | 136.689 | 0.0000* |
| Specification 5 | 137.015 | 0.0000* |
| Specification 6 | 132.486 | 0.0000* |

* and ** show significance levels at 1% and 5%, respectively.

Source: Authors' calculations

## 4.3. Discussion

The results of Tables 8 and 9 show the GLS results for the direct impact of globalization on employment creation. The results imply that out of 6 specifications, four specifications (1,2,4, and 5) point toward globalization's negative and significant impact on employment creation. In contrast, specification 3 shows negative and insignificant, and specification 6 shows globalization's positive and insignificant impact on employment creation. In four specifications

**Table 8. Direct impact of globalization on employment (GLS).**

| Dependent Variable: Employment as a Percentage of Population | | | | | | |
|---|---|---|---|---|---|---|
| | 1 | 2 | 3 | 4 | 5 | 6 |
| IVA | .1780909* (0.003) | .176598* (0.003) | .1695656* (0.005) | .1624321* (0.005) | .1409148** (0.014) | .1764832* (0.003) |
| EG | 2.360807* (0.000) | 1.704438* (0.001) | 3.337164* (0.000) | 2.094037* (0.000) | 2.417546* (0.000) | 2.726322* (0.000) |
| POP | -1.693679* (0.000) | -1.062661** (0.026) | -2.466991* (0.000) | -.39001 (0.434) | -.2768158 (0.539) | -1.81666* (0.000) |
| GLOB1 | .1299791* (0.035) | | | | | |
| GLOB2 | | .2217666* (0.000) | | | | |
| GLOB3 | | | -.0582014 (0.362) | | | |
| GLOB4 | | | | .2824274* (0.000) | | |
| GLOB5 | | | | | .2834219* (0.000) | |
| GLOB6 | | | | | | .0738602 (0.122) |
| Constant | 14.83404** (0.025) | 15.77721** (0.016) | 14.49869** (0.029) | -8.485026 (0.260) | -18.29157** (0.015) | 11.19542*** (0.108) |

*, ** and *** show significance levels at 1%, 5% and 10%, respectively.

Source: Authors' calculations

(1,2,4 and 5), 1 percent increase in globalization increases employment creation by 0.129%, 0.221%, 0.282% and 0.283%, respectively. In specifications 3 and 6, a 1 percent increase in globalization decreases and increases employment creation by 0.058% and 0.073%, respectively. However, this impact is insignificant.

Industrial value-added has a positive and significant impact on all specifications. In all specifications, 1 percent increase in industrial value-added increases employment creation by 0.178%, 0.176%, 0.169%, 0.162%, 0.140% and 0.176%, respectively. Economic growth has a positive and significant impact on all specifications. In all specifications, 1 percent increase in economic growth increases employment creation by 2.360%, 1.704%, 3.337%, 2.094%, 2.417% and 2.726%, respectively. Population growth negatively and significantly impacts employment creation in all specifications. In all specifications, 1 percent increase in population growth decreases employment creation by -1.693%, -1.062%, -2.466%, -0.390%, -0.276% and -1.816%, respectively. However, the coefficients in specifications 4 and 5 are insignificant. The sign of coefficients of these three variables are according to economic theory as the increase in

**Table 9. Indirect impact of globalization on employment through human capital (GLS).**

| Dependent Variable: Employment as a Percentage of Population | | | | | | |
|---|---|---|---|---|---|---|
| | 7 | 8 | 9 | 10 | 11 | 12 |
| IVA | .1744494* (0.004) | .1763915* (0.003) | .1714253* (0.004) | .1719126* (0.004) | .1634347* (0.006) | .1746687* (0.004) |
| EG | 2.851063* (0.000) | 2.419313* (0.000) | 3.259705* (0.000) | 2.251111* (0.000) | 2.204652* (0.000) | 2.774098* (0.000) |
| POP | -2.069258* (0.000) | -1.645982* (0.001) | -2.445865* (0.000) | -1.07278** (0.047) | -.6811601* (0.181) | -1.920309* (0.000) |
| GLOB1*HC | .0082655 (0.513) | | | | | |
| Glob2*HC | | .0238213** (0.055) | | | | |
| Glob3*HC | | | -.0079662 (0.513) | | | |
| Glob4*HC | | | | .0425012* (0.001) | | |
| Glob5*HC | | | | | .0555757* (0.000) | |
| Glob6*HC | | | | | | .0127799 (0.319) |
| Constant | 15.25878** (0.024) | 16.80293** (0.013) | 13.76042** (0.041) | 8.851696 (0.194) | 1.722426 (0.806) | 14.10125** (0.034) |

* and ** show significance levels at 1% and 5%, respectively.

Source: Authors' calculations

industrial growth and economic growth are a source of creating more employment avenues. In contrast, population growth becomes a hurdle to providing more employment opportunities for the increasing population.

The moderating role of human capital in the globalization-employment nexus has been estimated by the GLS model, as shown in Table 9. The three specifications (2, 4, and 5) point toward globalization's positive and significant impact on employment creation, whereas the two specifications (1 and 6) are positive and insignificant. One specification (3) shows globalization's negative and insignificant impact on employment creation.

Industrial value-added has a positive and significant impact on all specifications. In all specifications, 1 percent increase in industrial value-added increases employment creation by 0.174%, 0.176%, 0.171%, 0.171%, 0.163% and 0.174%, respectively. Economic growth has a positive and significant impact on all specifications. In all specifications, 1 percent increase in economic growth increases employment creation by 2.851%, 2.419%, 3.259%, 2.251%, 2.204% and 2.774%, respectively. Population growth negatively and significantly impacts employment creation in all specifications. In all specifications, 1 percent increase in population growth decreases employment creation by -2.069%, -1.645%, -2.445%, -1.072%, -0.681% and -1.920%, respectively. The sign of coefficients of these three variables are according to economic theory as the increase in industrial growth and economic growth are a source of creating more employment avenues. In contrast, population growth becomes a hurdle to providing more employment opportunities for an increasing population. The improvement in the results by adding a moderator of human capital in the model can be shown in the significance of population growth as it is significant in all specifications. The findings of the current study support the findings of [48–50].

Industry appears to be the most fantastic option for employment growth [59]. The industry continues to be a crucial source of substantial economic expansion and the corresponding generation of jobs, both directly and indirectly. First, some industries require much cash. The purpose of these sectors in terms of employment is less to create employment and more to permit higher employment in medium and high labor-intensive industries by delivering intermediate inputs at competitive costs. The second category of industries includes those in which capital and labor are complementary rather than substitutes. There is no contradiction between growing levels of fixed investment and job intensity in industries such as metals and plastics production or capital and transportation equipment—employment grows as capital investment rises. A fast-expanding industrial sector can also contribute significantly to indirect employment. Growth in manufacturing and the notable role of promoting backward linkages to primary industries encourage employment [73].

Economic growth can be a source of more employment generation. Higher growth leads to higher aggregate demand and supply and more employment opportunities. Many studies have analyzed multivariate and bivariate analyses between economic growth and employment [32, 74, 75].

As far as the impact of supporting variables on employment creation is concerned, the results in the static analysis show that industry value added and economic growth have a positive and significant impact on employment in all specifications. The impact of population growth on employment creation is negative, supporting the economic theory.

The empirical results showing the impact of globalization on employment are mixed in different specifications of static analysis, and nothing can be concluded about how globalization affects employment creation directly or indirectly through the channel of human capital. This points out that static analysis through GLS is not appropriate for empirical model analysis. Therefore, we employ GMM for the dynamic analysis of the model.

**Table 10. Direct impact of globalization on employment (system GMM results).**

| Dependent Variable: Employment as a Percentage of Population | | | | | | |
|---|---|---|---|---|---|---|
| | 1 | 2 | 3 | 4 | 5 | 6 |
| LEmp | .997372* (0.000) | .9961945* (0.000) | .9988119* (0.000) | .9940976* (0.000) | .9926056* (0.000) | .9972407* (0.000) |
| IVA | .0192015* (0.000) | .0192549* (0.000) | .0190227* (0.000) | .0190791* (0.000) | .0189302* (0.000) | .0191393* (0.000) |
| EG | .2579181* 0.005) | .2840525* (0.001) | .2927644* (0.001) | .3271127* (0.000) | .3364714* (0.000) | .3462478* (0.000) |
| PoP | -.5099753* 0.000) | -.5296593* (0.000) | -.5373264* (0.000) | -.5110491* (0.000) | -.5118545* (0.000) | -.5680713* (0.000) |
| Glob1 | .013118 (0.202) | | | | | |
| Glob2 | | .0091094 (0.286) | | | | |
| Glob3 | | | .0090111 (0.385) | | | |
| Glob4 | | | | .0128395 (0.112) | | |
| Glob5 | | | | | .0116053*** (0.065) | |
| Glob6 | | | | | | .0036517 (0.572) |
| Constant | 1.518111 (0.393) | 1.512573 (0.398) | 1.245993 (0.481) | .0246947 (0.990) | -.0361322 (0.984) | .8470123 (0.640) |

*, ** and *** show level of significance at 1%, 5% and 10%, respectively.

Source: Authors' calculations

In Table 10, we have estimated the model with the help of the generalized method of moments (GMM). All previous specifications of static analysis have been re-estimated. The results of dynamic analysis are different from the results of static analysis. The impact of lagged explained variable is positive and significant in all specifications regressions (1–6).

The coefficients of lagged employment show that 1 percent increase in previous year valued of employment will increase current year employment by 0.997%, 0.996%, 0.998%, 0.994%, 0.992% and 0.997%, respectively.

Industrial value added in all specifications positively and significantly affects employment creation. In all specifications, 1 percent increase in industrial value-added leads to 0.0192%, 0.0192%, 0.0190%, 0.0190%, 0.0189% and 0.0191% increase in employment creation, respectively.

Economic growth has a positive and significant impact on all specifications. In all specifications, 1 percent increase in economic growth increases employment creation by 0.257%, 0.284%, 0.292%, 0.327%, 0.336% and 0.346%, respectively. Population growth negatively and significantly impacts employment creation in all specifications. In all specifications, 1 percent increase in population growth decreases employment creation by 0.509%, 0.529%, 0.537%, 0.511%, 0.511% and 0.568%, respectively. According to economic theory, the sign of coefficients of these three variables is an increase in industrial growth, and economic growth is a source of creating more employment avenues. In contrast, population growth becomes a hurdle to providing more employment opportunities for the increasing population.

The impact of globalization on employment creation is insignificant in 5 out of six specifications. This may be due to the reason suggested by Alfalih and Hadj [13], according to which human capital may be a channel through which globalization affects employment creation. In order to check that, the moderating role of human capital has been checked in Table 11 and estimated by GMM to check the dynamic effects.

The results of Table 11 show that all variables have the same signs as in Table 10. However, an interesting finding is that the moderating variable of globalization and human capital is positive and significant in five of six specifications. Table 10 was significant in only one specification. This points toward the finding that the direct impact of globalization on employment creation is insignificant. In contrast, globalization creates employment opportunities in host

**Table 11. Indirect impact of globalization on employment through human capital (system GMM results).**

| Dependent Variable: Employment as a Percentage of Population | | | | | | |
|---|---|---|---|---|---|---|
| | 1 | 2 | 3 | 4 | 5 | 6 |
| LEmp | .9972534* (0.000) | .9963677* (0.000) | .9986572* (0.000) | .995553* (0.000) | .9953052* (0.000) | .9968687* (0.000) |
| IVA | .0196882* (0.000) | .019765* (0.000) | .0195201* (0.000) | .0197573** (0.000) | .0194154* (0.000) | .0194829* (0.000) |
| EG | .2391813* (0.004) | .2512676* (0.002) | .2397964* (0.003) | .2866948* (0.000) | .299296* (0.000) | .3058608* (0.000) |
| POP | -.4902849* (0.000) | -.5000081* (0.000) | -.4898072* (0.000) | -.4838547* (0.000) | -.4888607* (0.000) | -.5222434* (0.000) |
| GLOB1*HC | .0037945*** (0.087) | | | | | |
| GLOB2*HC | | .0033264*** (0.107) | | | | |
| GLOB3*HC | | | .0039335*** (0.077) | | | |
| GLOB4*HC | | | | .0040996** (0.051) | | |
| GLOB5*HC | | | | | .0035946** (0.036) | |
| GLOB6*HC | | | | | | .0028117 (0.165) |
| Constant | 1.879127 (0.297) | 1.872524 (0.302) | 1.73865 (0.330) | .6497773 (0.714) | .5087845 (0.774) | .9201228 (0.602) |

*, ** and *** show significance levels at 1%, 5% and 10%, respectively.

Source: Authors' calculations

countries if they have sufficient human capital. In all specifications, 1 percent increase in interaction term (GLOB*HC) increases employment creation by 0.0037%, 0.0032%, 0.0039%, 0.0040%, 0.0035% and 0.002%, respectively. However, the coefficient is positive but insignificant. The current study's findings support the findings of previous studies by [13, 30, 43–49]. The findings contradict [40].

## 5. Conclusion

This study analyzes the moderating role of human capital in the globalization-employment nexus in twenty-six Asian countries. For this, annual panel data has been collected from 1996 to 2019 using 12 specifications, i.e., six specifications for direct. In contrast, six specifications check the indirect impact of globalization on employment creation through the human capital channel. The heteroscedasticity and serial correlation have been tested by employing the Wald test and Wooldridge test, respectively, which show the presence of both problems. Therefore, the static relationship between variables has been checked using GLS technique. The GMM technique has checked the dynamic relationship.

The results of static analysis by GLS show the direct impact of globalization on employment creation. The results imply that out of 6 specifications, four specifications (1,2,4, and 5) point toward globalization's negative and significant impact on employment creation. In contrast, specification 3 shows negative and insignificant, and specification 6 shows globalization's positive and insignificant impact on employment creation. Therefore, the moderating role of human capital in the globalization-employment nexus has been estimated by the GLS model. The three specifications (2, 4, and 5) point toward a positive and significant impact of globalization on employment creation, whereas the two specifications (1 and 6) are positive and insignificant. One specification (3) shows globalization's negative and insignificant impact on employment creation in the presence of human capital.

The static analysis shows mixed results, which suggests applying dynamic analysis through GMM. All previous specifications of static analysis have been re-estimated. The results of dynamic analysis are different from the results of static analysis. The impact of lagged explained variable is positive and significant in all specifications. The impact of globalization on employment creation is insignificant in 5 out of six specifications. As suggested by Alfalih

and Hadj [13], globalization affects employment creation through the channel of human capital. In order to check that, the moderating role of human capital has been checked. The moderating variable of globalization and human capital is positive and significant in five out of six specifications compared to only one indirect analysis specification. This points toward the finding that the direct impact of globalization on employment creation is insignificant in most specifications. In contrast, globalization creates employment opportunities in host countries if they have a sufficient level of human capital, as recently found by Alfalih and Hadj [13].

Overall, industrial value added, and economic growth are positively and significantly associated with employment creation. In contrast, population growth negatively and significantly impacts employment creation in all specifications. The sign of coefficients of these three variables are according to economic theory as the increase in industrial growth and economic growth are a source of creating more employment avenues. In contrast, population growth becomes a hurdle to providing more employment opportunities for an increasing population.

### 5.1. Policy recommendation

Regarding where the field of study in human capital development is headed, the latest research has included the relevant cultural, psychological, and social aspects in the theoretical frameworks [76, 77]. As human capital plays a positive role in getting the advantage of globalization [78], we assert that the economies analyzed here should put more emphasis on human capital if they want to increase their capacity to generate new employment. This will enable them to achieve higher employment in time of globalization.

### 5.2. Future research

The limitation of the current study is that it considers only Asian economies. The reason for this is that these economies have a matching level of human capital as well as employment structures. Future research may consider the comparative analysis of developing and developed economies.

## Supporting information

**S1 Appendix.**
(DOCX)

## Author Contributions

**Conceptualization:** Mansoor Mushtaq, Shabbir Ahmed.

**Data curation:** Mansoor Mushtaq, Shabbir Ahmed.

**Formal analysis:** Mansoor Mushtaq, Shabbir Ahmed.

**Funding acquisition:** Shabbir Ahmed.

**Investigation:** Mansoor Mushtaq, Shabbir Ahmed, Mochammad Fahlevi.

**Methodology:** Mansoor Mushtaq, Shabbir Ahmed, Mochammad Fahlevi.

**Project administration:** Mochammad Fahlevi.

**Resources:** Mochammad Fahlevi, Mohammed Aljuaid.

**Software:** Mochammad Fahlevi, Mohammed Aljuaid.

**Supervision:** Mohammed Aljuaid, Sebastian Saniuk.

**Validation:** Mohammed Aljuaid, Sebastian Saniuk.

**Visualization:** Mohammed Aljuaid, Sebastian Saniuk.

**Writing – review & editing:** Sebastian Saniuk.

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
