## [Decision Letter · Decision Letter 0]

26 Aug 2022

PONE-D-22-21757Globalization and Employment Nexus: Moderating Role of Human CapitalPLOS ONE

Dear Dr. FAHLEVI,

Thank you for submitting your manuscript to PLOS ONE. After careful consideration, we feel that it has merit but does not fully meet PLOS ONE’s publication criteria as it currently stands. Therefore, we invite you to submit a revised version of the manuscript that addresses the points raised during the review process.

We look forward to receiving your revised manuscript.

Kind regards,

Jabbar Ul-Haq, PhD

Academic Editor

PLOS ONE

Journal Requirements:

Reviewers' comments:

Reviewer's Responses to Questions

**Comments to the Author**

1. Is the manuscript technically sound, and do the data support the conclusions?

Reviewer #1: Yes

Reviewer #2: Yes

Reviewer #3: Partly

Reviewer #4: Yes

2. Has the statistical analysis been performed appropriately and rigorously? 

Reviewer #1: Yes

Reviewer #2: No

Reviewer #3: Yes

Reviewer #4: Yes

3. Have the authors made all data underlying the findings in their manuscript fully available?

Reviewer #1: Yes

Reviewer #2: Yes

Reviewer #3: No

Reviewer #4: Yes

4. Is the manuscript presented in an intelligible fashion and written in standard English?

Reviewer #1: Yes

Reviewer #2: Yes

Reviewer #3: Yes

Reviewer #4: Yes

5. Review Comments to the Author

Reviewer #1: This paper is interesting to deal with globalization and employment, which is the most important due to COVID-19 consequences. Therefore, before made a final decision, it should be further improved.

Abstract.

This part should be restructured and consciously.

Introduction :

This part is failed to explain the research questions, problem and significant of the study. Besides, there is no valid reason of choosing the Asian economies. I strongly suggest that clearly describe the strong reason of choosing the Asian economies as a sample. Globalization does not only affect the one region but overall the world.

Literature Review:

I suggest that literature review part should be described in the sub-section e.g., nexus between globalization and employment etc. Please, re-arrange according to specific segments.

Empirical and discussion :

These parts should be emerged into one and concisely described with mechanical and theoretical interpretations. The drawback of this part is, authors explained only mechanical interpretation and very less. I strongly suggest that please describe all results with critically and compare with former studies.

Also, author did not define the term' specification 1-6' in the whole paper but use the model. Please clearly define and argue why use ?

Conclusion:

I saw that authors explained 6 specifications in the model but they are talking about 12 specification. Please clarify with arguments.

This study analyzes the moderating role of human capital in globalization-employment nexus in

30 Asian countries. For this, annual panel data has been collected from 1996 to 2019 by using 12

specifications. The static relationship between variable has been checked by using fixed effects

model and random effects model techniques. The dynamic relationship has been checked by GMM

approach.

This part should be extended more and please provide policy recommendations based on empirical findings in separate section.

Good luck

Reviewer #2: The main aim of this paper is to investigate analyzes the moderating role of human capital in globalization employment nexus in 26 Asian countries. The topic is interesting, and it is a good idea to investigate this role.

The empirical literature has been presented in a systematic manner but needs moe up to date citations. The discussion of the results is fine too. However, the FE and RE results maybe compared with the results of the system GMM if only the initial value of employment is included in these specifications. Also, the sentence structure must be improved and some grammatical errors removed in order to allow for the logical flow of ideas. I have pointed out some of these issues below. Furthermore, there is no policy implication and recommendations for further study. If all these concerns are taken care of, the paper will be more interesting.

Reviewer #3: Manuscript number: PONE-D-22-21757

Title: Globalization and Employment Nexus: Moderating Role of Human Capital

General comment

This work is interesting work that needs major revision to be fit for publications in POLS ONE.

Recommendations: Major revision

1. Abstract

Abstract should start with problem statement and conclude with policy recommendations briefly. The organization of abstract is not satisfactory

2. Introduction

Introduction is not satisfactory authors should start with research problem along with providing detail rationale of the study. The novelty factor should be also highlighted properly.

3. Literature Review

The literature review section should discuss with theoretical model and discuss critically existing literature to highlight gap that this study aims to address. This section should explain novelty contribution of the study.

4. Methodology

The author should highlight source of equations, explain rationale why authors have used so many equations. Second, with test authors used to check multicollinearity problem. Also explain the rationale of variable used along with expected sign.

5. Results and Discussion

A real discussion of the findings is lacking. The discussion of empirical results indicates the lack of knowledge in your study area. Discussion and implications need to underline the contributions along with support from prior literature. Moreover, the diagnostics test analysis is missing here.

6. Conclusion

Conclusion part should highlight findings of this study that will lead to policy implications. Moreover, limitation of study and direction for future research should be properly highlighted.

Reviewer #4: Manuscript number: PONE-D-22-21757

Title: Globalization and Employment Nexus: Moderating Role of Human Capital

General comment:

This study investigates the Globalization and Employment Nexus: Moderating Role of Human Capital using annual panel datasets of Asian economies. This paper has some contribution to the literature. However, there are a lot of problems. Thus, I think this paper need major revision before publish in PLOS ONE

Recommendations: Major revision

Abstract

• The abstract need to comprehensively improve.

• Study should introduce policy implications.

• Need to add jell code.

Introduction

• Introduction of the study is poor so need improvement and rewrite due to lack of knowledge.

• The contribution of the study is ambiguous, the study should improve the contribution and rewrite.

• Study need to give support by citing references, “Globalization is defined as the combination of capital flows, foreign direct investment, trade, technology, and labor mobility with the national economy” , “On the influence of globalization on employment, there is no consistency among researchers”

• Study needs to cite the updated references

Literature Review

• Study should describe the seminal work about the association between globalization and employment.

• The study literature review is too weak and need to update till 2022

• Novelty of the study is not sufficiently clarify, the paper did highlight the contribution of the paper in the literature, but it is quite weak. It is important to identify the weakness of previous papers, and show the argument and contribution of paper.

• Remove the heading of novelty from literature section and add the literature summary.

Theoretical Framework

• The study need to describe the theoretical relationship between globalization, employment, and human capital based on some theory.

• The study should add the theoretical framework and theoretical channels about how the variable affect each other.

Data and Methodology

• The sample period is from 1996-2019, need to update the dataset as it is 2022.

• The paper should include the variables description table.

• Provide the reference from previous literature who used the same proxy of variables and include expected sign for the variables used in study.

Results and Discussion

• A real discussion of the findings is lacking. The discussion of empirical results indicates the lack of knowledge in your study area.

• The study should interpret results in results section (with tables).

• The length of the paper seems to be inappropriate. It should be more comprehensive.

• Discussion and implications need to underline the contributions of the paper by hooking them to previous research.

• Moreover, the diagnostics test analysis is missing here.

• Check the first sentence of econometric analysis. “As we used panel data therefore we have to decide whether fixed effect or random effect model should be used”

• Need to check tables as some models are insignificant and justify the estimated sign in discussion section.

• The study support the findings with previous literature (The findings of the current study support the findings of previous studies of Lee et al. (2010), Malik et al., (2011), Gozgor (2014), Awad and Youssof (2016), Daly et al. (2017), Gozgor (2017), Adamu et al. (2018), Siddiqa et al. (2018) and Alfalih and Hadj (2021).), also need here to contradict the findings with previous literature.

• Add zero (0) before point in models.

Conclusion

• Conclusion need to make attractive by adding comprehensive detail and add some suitable starting paragraph/sentence by following close literature to your study.

• Conclusion and recommendation should be written as a separate headings

• The sphere of impact of the research and considerable aspect of furtherance of the study should be stated.

6. PLOS authors have the option to publish the peer review history of their article (what does this mean?). If published, this will include your full peer review and any attached files.

Reviewer #1: No

Reviewer #2: No

Reviewer #3: **Yes: **Shujaat Abbas

Reviewer #4: **Yes: **Sana Khanum

---

## [Author Response · Author response to Decision Letter 0]

9 Sep 2022

Greetings, Editor

We appreciate your letting us submit a revision. Our anonymous expert's remarks were completely genuine, and we were happy to make the improvements they recommended.

We believe these changes in these revisions addressed everything necessary that was expected, but we want to reassure you that any new suggestions our experts may have would be welcomed. 

The letter with our response and each item addressed, in turn, can be found below. In the manuscript text, we have color-coded both experts and provided numbers next to each of their comments to make things easier to comprehend.

Looking forward to starting revisions process

Regards,

Corresponding Author

Reviewer 1 Changings Recommended and reply

Received Comments 1. The paper has not included a theoretical framework describing the link between globalization and employment and how the other variables might affect employment.

Reply 1. We appreciate our reviewer's thoughtful comments. You can track all of the red-highlighted points you made for us. As a result of answering point 1, we have provided the necessary theoretical framework.

Received Comments 2. In the abstract, the authors stated that annual were collected from 1996 to 2019 using 12 specifications. How do we use specifications in the collection of data? Can they explain how a specification can be used to collect data? I suppose the authors meant the collected data were using in 12 specifications.

Reply 2. We appreciate you making the remark and highlighting the problem you believe needs to be fixed. Correction has been made accordingly

Received Comments 3. In lines 1 and 2 on page 8(introduction), Globalization is defined as the combination of capital flows, foreign direct investment, trade, technology, and labor mobility with the national economy. What does it mean by “with the economy? Or better still, they should include a citation.

Reply 3. Thank you for your valuable point, this explanation was necessary. Correction has been made and citation has been included as you indicated

Received Comments 4. In line 34 to 36 of page 9, the phase “The first of these takes into account the trade liberalization and the second foreign direct investment as the main indicators of economic globalization” should be rephrased to make more sense.

Reply 4. thank you for identifying necessary corrections and need of sentences to be rephrased. Hence compliance has been made accordingly

Received Comments 5. As far as the methodology is concerned, the authors have focused on the Hausman specification test and have not check the issues of heteroscedasticity, cross sectional dependence and serial correlation, if these issues are identified, the PCSE and GLS techniques are most appropriate techniques. Also, FE with Driscoll-Kraay standard errors can be used. These are completely absent.

Reply 5. Thank you for indicting a better technique for this paper. Please follow our comments: Wald test for heteroscedasticity and Wooldridge test have been employed which pointed to presence of both problems. Therefore, generalized least squares (GLS) method has been applied for empirical analysis.

Received Comments 6. In Lines 4 to 6 of the discussion paragraph on page 20, the statement “As far as indirect impact of globalization on employment creation through human capital via fixed effects model is in table 5 is concerned” is not understood. The authors have to fix this.

Reply 6. Thank you for your emphasis, which may have reduced comprehension. Hence correction has been made accordingly.

Received Comments 7. The statement ‘However, the impact of economic growth on negative on employment creation” in lines 22 to 23 of page 20 has to be revisited.

Reply 7. The correction has been made considering your points in mind; we hope you will see them compellingly explaining the intended meaning.

Received Comments 8. The policy implication and recommendations for further studies are absent.

Reply 8. Thank you for highlighting shortcoming. As a result. We have provided Policy implication and recommendations for further studies considering recommendations in mind.

We appreciate your thoughtful comments and Major and minor insights. We think this paper is a better version than the previous one, and we appreciate your guidance in making the manuscript better.

Reviewer 2 Changings Recommended and author reply 

Received Comments 1. Abstract should start with problem statement and conclude with policy recommendations briefly. The organization of abstract is not satisfactory 

Reply 1. We appreciate your feedback and agree that your proposal is sound, thus we did what you suggested.

Received Comments 2. Introduction is not satisfactory authors should start with research problem along with providing detail rationale of the study. The novelty factor should be also highlighted properly. 

Reply 2. Our entire team of coauthors concurred on this point, and it is appropriate to act as intended thus we followed your recommendation. We hope that these changes will be convincing enough for our expert requirements.

Received Comments 3. The literature review section should discuss with theoretical model and discuss critically existing literature to highlight gap that this study aims to address. This section should explain novelty contribution of the study. 

Reply 3. This comment was taken into consideration and Corrections have been made as required. We will be waiting for your feedback.

Received Comments 4. The author should highlight source of equations, explain rationale why authors have used so many equations. Second, with test authors used to check multicollinearity problem. Also explain the rationale of variable used along with expected sign. 

Reply 4. the recommendations have been followed; we will be looking forward for the feedback.

Received Comments 5. A real discussion of the findings is lacking. The discussion of empirical results indicates the lack of knowledge in your study area. Discussion and implications need to underline the contributions along with support from prior literature. Moreover, the diagnostics test analysis is missing here.

Reply 5. Thank you for your comments. We tried to overcome flaws you noticed and followed the recommendations have made. We expect that our efforts will be in accordance with what our reviewer anticipates.

Received Comments 6. Conclusion part should highlight findings of this study that will lead to policy implications. Moreover, limitation of study and direction for future research should be properly highlighted. 

Reply 6. We see this point was equally important and highlighted by our other reviewer as well. We thank you for emphasizing for the need of policy implications. Findings of study, policy implications and direction of future research have been included.

We value your insightful feedback and all of your insights. We believe that this work is an improvement over the previous one, and we value your suggestions for making the manuscript stronger.

---

## [Decision Letter · Decision Letter 1]

29 Sep 2022

PONE-D-22-21757R1Globalization and Employment Nexus: Moderating Role of Human CapitalPLOS ONE

Dear Dr. FAHLEVI,

Thank you for submitting your manuscript to PLOS ONE. After careful consideration, we feel that it has merit but does not fully meet PLOS ONE’s publication criteria as it currently stands. Therefore, we invite you to submit a revised version of the manuscript that addresses the points raised during the review process.

We look forward to receiving your revised manuscript.

Kind regards,

Jabbar Ul-Haq, PhD

Academic Editor

PLOS ONE

Journal Requirements:

Additional Editor Comments:

Authors are required to respond to Reviewer 2 comments only

Reviewers' comments:

Reviewer's Responses to Questions

**Comments to the Author**

1. If the authors have adequately addressed your comments raised in a previous round of review and you feel that this manuscript is now acceptable for publication, you may indicate that here to bypass the “Comments to the Author” section, enter your conflict of interest statement in the “Confidential to Editor” section, and submit your "Accept" recommendation.

Reviewer #1: All comments have been addressed

Reviewer #2: (No Response)

Reviewer #3: All comments have been addressed

Reviewer #4: All comments have been addressed

2. Is the manuscript technically sound, and do the data support the conclusions?

Reviewer #1: Yes

Reviewer #2: Yes

Reviewer #3: Yes

Reviewer #4: Yes

3. Has the statistical analysis been performed appropriately and rigorously? 

Reviewer #1: Yes

Reviewer #2: Yes

Reviewer #3: Yes

Reviewer #4: Yes

4. Have the authors made all data underlying the findings in their manuscript fully available?

Reviewer #1: Yes

Reviewer #2: Yes

Reviewer #3: Yes

Reviewer #4: Yes

5. Is the manuscript presented in an intelligible fashion and written in standard English?

Reviewer #1: Yes

Reviewer #2: Yes

Reviewer #3: Yes

Reviewer #4: Yes

6. Review Comments to the Author

Reviewer #1: Dear authors, thank you for incorporating the issues in the final draft. Thus, I would like to accept the draft in its current form.

Reviewer #2: After carefully reading the paper, I can now confirm that the authors addressed all eight of the initial concerns. This revised manuscript satisfies the requirements for publication in PLUS ONE Journal. As a result, I believe the paper should be considered for publication in your esteemed journal.

The authors must, however, address the following minor concerns:

1. In the theoretical framework, the sentence “According to the Heckscher-Ohlin model, globalization increased employment in developing countries” should be According to the Heckscher-Ohlin model, globalization will increase employment in developing countries.

2. “Are” should be deleted from this sentence “However, the diagnostic tests are shown in tables 4 to 7 point out that the model suffers from heteroscedasticity and serial correlation problems”.

Reviewer #3: All comments incorporated by the authors. Therefore I am accepting this article for publication. Good Luck

Reviewer #4: This paper has some contribution to the literature and the authors have addressed all comments in a good way and may be accepted for publication on the Globalization and Employment Nexus: Moderating Role of Human Capital.

7. PLOS authors have the option to publish the peer review history of their article (what does this mean?). If published, this will include your full peer review and any attached files.

Reviewer #1: No

Reviewer #2: No

Reviewer #3: No

Reviewer #4: **Yes: **Sana Khanum

---

## [Author Response · Author response to Decision Letter 1]

3 Oct 2022

1. We appreciate our reviewer's thoughtful comments. We change to “According to the Heckscher-Ohlin model, globalization will increase employment in developing countries”

2. We appreciate you making the remark and highlighting the problem you believe needs to be fixed. Correction has been made accordingly, we change to “However, the diagnostic tests shown in tables 4 to 7 point out that the model suffers from heteroscedasticity and serial correlation problems”

---

## [Editor Report · Decision Letter 2]

7 Oct 2022

Globalization and Employment Nexus: Moderating Role of Human Capital

PONE-D-22-21757R2

Dear Dr. FAHLEVI,

We’re pleased to inform you that your manuscript has been judged scientifically suitable for publication and will be formally accepted for publication once it meets all outstanding technical requirements.

Kind regards,

Jabbar Ul-Haq, PhD

Academic Editor

PLOS ONE
---

## [Editor Report · Acceptance letter]

13 Oct 2022

PONE-D-22-21757R2 

Globalization and Employment Nexus: Moderating Role of Human Capital 

Dear Dr. FAHLEVI:

I'm pleased to inform you that your manuscript has been deemed suitable for publication in PLOS ONE. Congratulations! Your manuscript is now with our production department. 

Kind regards, 

on behalf of

Dr. Jabbar Ul-Haq 

Academic Editor

PLOS ONE